# Sequence-based virtual screening using transformers

**Shengyu Zhang** [1,3], **Donghui Huo**[1,2,3], **Robert I. Horne** [1,3], **Yumeng Qi**[1,3], **Sebastian Pujalte Ojeda**[1], **Aixia Yan**[2] **& Michele Vendruscolo** [1] ✉

Protein-ligand interactions play central roles in myriad biological processes and are of key importance in drug design. Deep learning approaches are becoming cost-effective alternatives to high-throughput experimental methods for ligand identification. Here, to predict the binding affinity between proteins and small molecules, we introduce Ligand-Transformer, a deep learning method based on the transformer architecture. Ligand-Transformer implements a sequence-based approach, where the inputs are the amino acid sequence of the target protein and the topology of the small molecule to enable the prediction of the conformational space explored by the complex between the two. We apply Ligand-Transformer to screen and validate experimentally inhibitors targeting the mutant EGFR[LTC] kinase, identifying compounds with low nanomolar potency. We then use this approach to predict the conformational population shifts induced by known ABL kinase inhibitors, showing that sequence-based predictions enable the characterisation of the population shift upon binding. Overall, our results illustrate the potential of Ligand-Transformer to accurately predict the interactions of small molecules with proteins, including the binding affinity and the changes in the free energy landscapes upon binding, thus uncovering molecular mechanisms and facilitating the initial steps in drug design.

Recent reports are revealing that deep learning approaches exhibit capabilities beyond their original function of predicting protein structures, including the possibility to infer protein dynamics and protein interactions[1–13]. These developments have opened the way to the use of deep learning in early-stage drug design[14–20] to help address the problem of the progressive inefficacy of drug discovery pipelines[21,22]. One important aspect of this problem is the identification of ligands using high-throughput screening, which is resource-intensive, requiring substantial financial investment, extensive time, and specialized equipment. Additionally, the process can be inefficient due to the high rate of false positives and negatives, necessitating further validation steps that add to the overall cost and time burden. These limitations have driven the development of computational methods

for drug design in the last several decades[23–26]. The advent of deep learning is now creating novel opportunities for more efficient, accurate, and cost-effective approaches to predict protein-ligand interactions by significantly reducing the experimental workload.

Here, we explore the idea of sequence-based drug design to use deep learning to predict how a protein target explores its conformational space when in complex with a candidate ligand. The goal with sequence-based drug design is to go beyond structure-based[18,23,25,26] and ligand-based[27,28] drug design methods, which largely rely on the conformations of the interaction partners in their free states. By taking as input the amino acid sequence of the protein target and the structure of the ligand, this approach offers as output a prediction of the conformational space of the complex and the corresponding binding

[1]Centre for Misfolding Diseases, Yusuf Hamied Department of Chemistry, University of Cambridge, Cambridge, UK. [2]College of Life Science and Technology, Beijing University of Chemical Technology, Beijing, China. [3]These authors contributed equally: Shengyu Zhang, Donghui Huo, Robert I. Horne, Yumeng Qi. ✉e-mail: mv245@cam.ac.uk

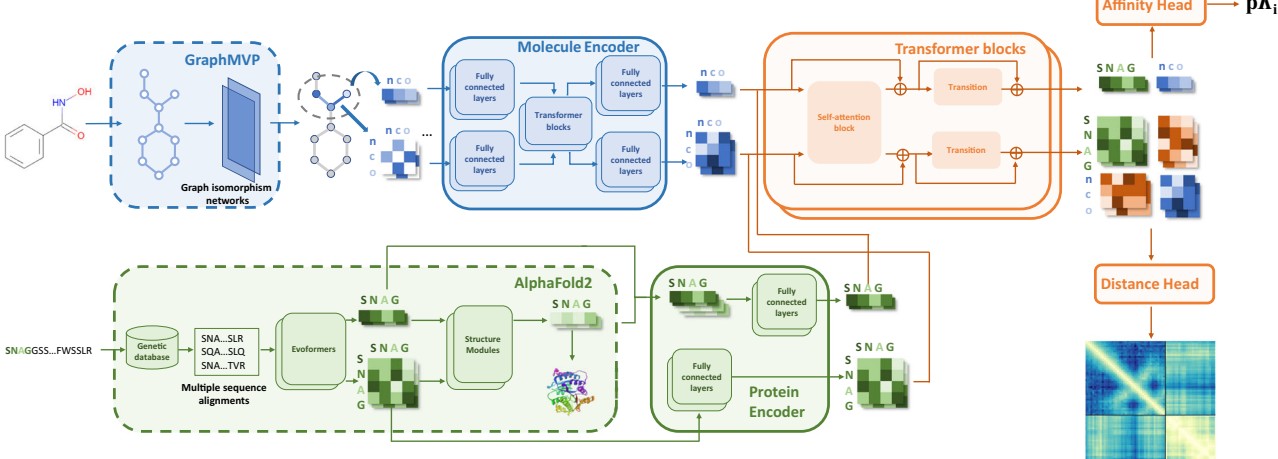

**Fig. 1 | Model architecture of Ligand-Transformer.** Ligand-Transformer represents a protein-ligand complex as a heterogeneous graph, incorporating residue and atom sets from the protein and ligand, respectively, along with pairwise features. The graph is formed from inputs generated by AlphaFold2 for proteins and GraphMVP for ligands, which are then re-encoded into an initial complete graph. This graph is subsequently refined through a 12-layer transformer-like network. This network updates both node and edge representations via self-attention with pair bias. The output is processed by the affinity head for binding affinity prediction and by the distance head for distance distribution prediction. Components within dotted-line boxes have fixed parameters, whereas those within solid lines are trainable.

affinity. We thus report Ligand-Transformer, a deep learning approach designed to model protein-ligand interactions.

The architecture of Ligand-Transformer is presented in Fig. 1. Briefly, Ligand-Transformer is based on the transformer framework of AlphaFold2[6] to generate protein representations from their sequences, and the Graph Multi-View Pre-training (GraphMVP) framework[29] to generate ligand representations. Instead of using the final predicted protein structure, we leverage the intermediate outputs. For the ligands, during pre-training, GraphMVP injects the knowledge of 3D molecular geometry into a 2D molecular graph encoder, allowing downstream tasks to benefit from the implicit 3D geometric prior. By leveraging the prior knowledge encoded in these high-dimensional representations, we capture the structural features of protein-ligand interactions, resulting in an accurate modelling of the bound states. The representations are further processed by the main structure of Ligand-Transformer, which consists of three parts (Fig. 1). The first part is the feature encoders to re-process the representations of proteins and ligands. The second part is the cross-modal attention network to exchange the information between representations of the protein and ligand. The third part is composed of two downstream predictors, the first head for affinity predictions and the second head for distance predictions.

Here, we describe Ligand-Transformer, a sequence-based deep learning framework that models protein-ligand interactions by predicting both the binding affinity and the conformational space of protein-ligand complexes. We apply it to identify inhibitors of the drug-resistant EGFR[LTC] kinase, achieving a hit rate of 58%, with two ligands exhibiting low-nanomolar affinity in validation binding experiments. We further show that Ligand-Transformer accurately predicts ligand-induced conformational population shifts of ABL kinase, consistent with experimentally determined conformational states. These results illustrate how Ligand-Transformer enables out-of-distribution predictions and captures the free energy landscape upon binding, offering a scalable and efficient alternative to high-throughput experimental assays in early-stage drug discovery.

## Results

### Performance comparison against state-of-the-art affinity prediction methods

We conducted a performance comparison of Ligand-Transformer against other affinity prediction methods[30–32] utilizing the PDBbind2020 dataset

(Table S1). Our results indicate that Ligand-Transformer achieves comparably better correlations with experimentally-measured values when compared to baseline methods.

To evaluate the predictive accuracy of Ligand-Transformer on both binding affinity and distance matrices, we used the PDBbind2020 dataset. We curated a subset of 13,420 complexes (Supplementary Data 1), ensuring manageable computational loads by limiting the maximum length of protein sequences to 384 residues, and the maximum number of atoms in each ligand to 128. Each complex in the dataset has an experimentally measured binding affinity ($pK_d$), allowing us to compare predicted values directly against the measured data. We randomly split the dataset into training (10,375 complexes), validation (640), and test (936).

To compare Ligand-Transformer to competing approaches, we also trained three other deep learning-based affinity prediction models[30–32] on the same data partitions. Table S1 and Fig. S1 summarize these comparisons, showing that Ligand-Transformer achieves higher or on-par correlation with experimentally measured affinities relative to all three baseline methods. Furthermore, Ligand-Transformer effectively predicts protein-ligand distances. We found that approximately 95% of the residue-residue distance errors were below 0.5 Å, while ~95% of the residue-ligand atom distance errors were within 2 Å (Fig. S2). These results suggest that Ligand-Transformer is able to capture structural aspects of protein-ligand complexes that are useful for both binding affinity and distance matrix predictions. Additional details of these comparisons, as well as the ability of the model to estimate its own error and to generalize to unseen protein-ligand combinations (Figs. S3 and S4), are provided in the "Methods" and the Supplementary Methods.

### Identification of EGFR[LTC] ligands

We illustrate the potential of Ligand-Transformer for the identification of initial hits in drug discovery pipelines by screening ligands targeting EGFR[LTC], a mutant form of the EGFR kinase. EGFR is a key target in cancer therapy due to its role in cell growth[33]. As mutations in EGFR, such as L858R/T790M/C797S (LTC), can lead to resistance against all current EGFR inhibitors, there is a need for novel drugs to target this triple-mutant[34]. We first collected a dataset, called EGFR[LTC]-290 (Supplementary Data 2), consisting of 290 existing inhibitors with their measured half maximal inhibitory concentration (IC$_{50}$) values, along

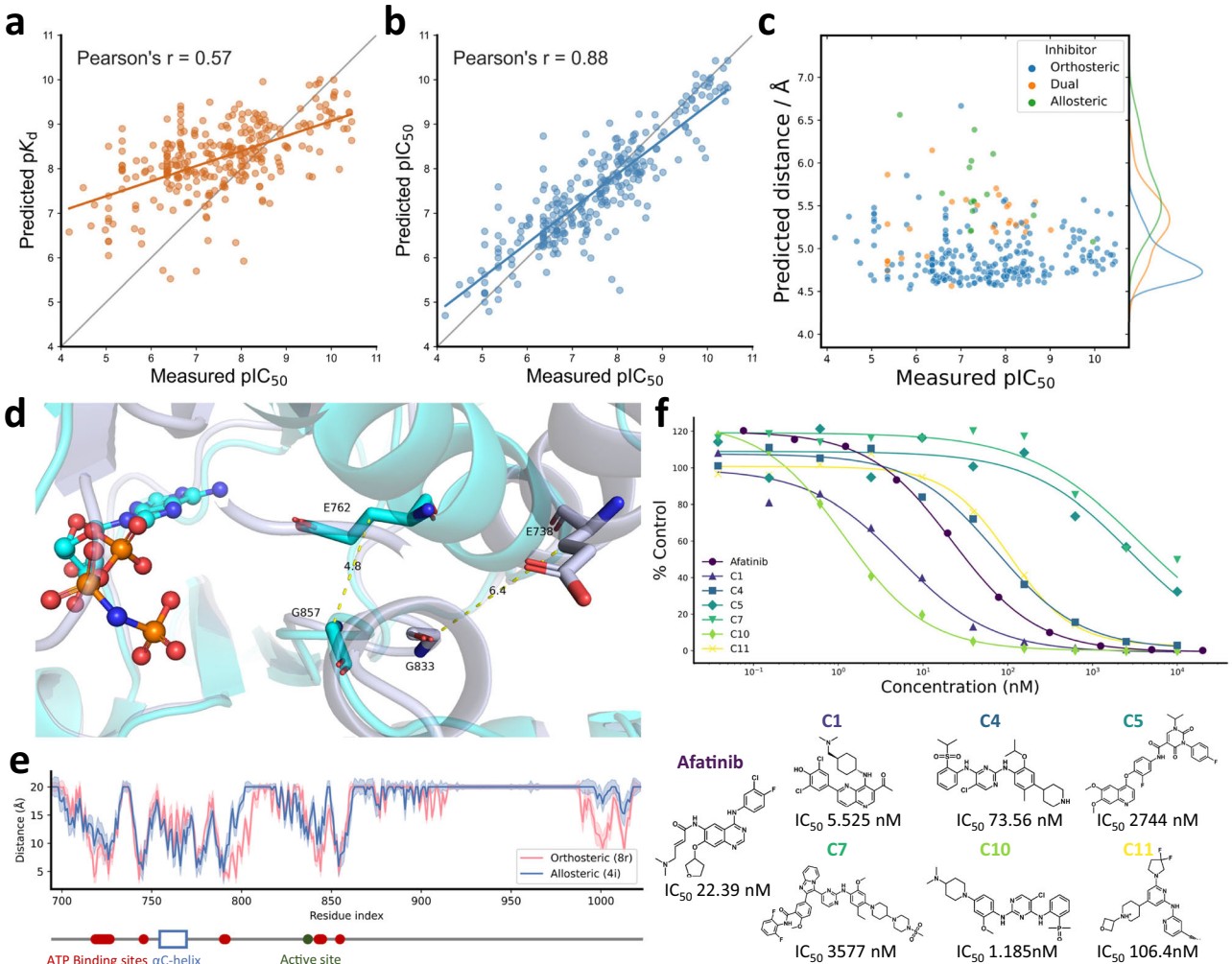

**Fig. 2 | Identification of ligands targeting EGFR^LTC. a, b** Correlation between the experimental pIC50 values and the binding parameters (p$K_d$ and pIC50 values) using Ligand-Transformer for complexes in the EGFR^LTC-290 dataset; all binding parameters are in molar units: **a** predicted p$K_d$ values without transfer learning (base model), and **b** predicted pIC50 values post tenfold validation with transfer learning. **c** Distribution of predicted distances between E762 and G857 of EGFR^LTC in various ligand-binding states. *y*-axis: predicted distance between the E762 Cβ and G857 Cα atoms; *x*-axis: experimental pIC50 values of the ligands. Data points are colored based on their reported binding sites in the literature: orthosteric (blue), allosteric (green), and dual (orange). The kernel density estimations (KDE) plots for the distributions of predicted distances are indicated adjacent to y coordinates. **d** Comparison of the EGFR kinase domain X-ray structures in the active (PDB ID 2ITX) and inactive (PDB ID 2GS7) states. The αC-helix is shown as a ribbon in light blue and gray, representing the active and inactive states, respectively. The AMP-PNP ligand bound to PDB ID 2ITX is visualized as a stick-and-ball structure. Residues

E762 and G857 of EGFR (corresponding to residues E738 and G833 in PDB ID 2GS7) are shown as sticks. Nitrogen, oxygen, phosphorus, and carbon atoms are colored dark blue, red, orange, and light blue, respectively. **e** Line graph of the predicted binding modes of an orthosteric inhibitor (8r[54], colored in red) and an allosteric inhibitor (4i[55], colored in blue) with EGFR^LTC. The line graph illustrates the predicted minimal distances between each residue and the ligand. The error bars represent the predicted confidence scores (pMAE, see **Algorithm 8** in Supplementary Information). The bottom part shows sequence annotation of EGFR based on UniProt P00533. ATP binding sites are shown in red dots, active site is shown in green dot, and the region of αC-helix is shown in blue box. **f** In vitro kinase inhibition assays for selected compounds, depicted as a percentage of activity relative to a DMSO control. The graph presents the mean of duplicate measurements for each ligand. The corresponding IC50 values are presented for each compound. Source data are provided as a Source Data file.

with annotations indicating whether they are allosteric or orthosteric (see "Methods"). We found that Ligand-Transformer could predict the binding affinity with a Pearson's correlation coefficient (R) value of 0.57 (Fig. 2a). To fine-tune the model on this specific dataset to achieve higher accuracy, we randomly split the EGFR^LTC-290 dataset into ten parts, and conducted a tenfold cross validation to evaluate the performance and obtain an ensemble of fine-tuned models (Model FT1 to FT10, see "Methods"). The value of R of this test increased to 0.88 after fine tuning (Fig. 2b).

We identified notable differences in the binding modes of the predicted orthosteric and allosteric inhibitors. Statistical analysis using the *t*-test revealed no significant differences in activity distribution between the orthosteric and allosteric groups (*p* = 0.66)

and the dual group (*p* = 0.39) (Fig. 2c). However, the predicted distance distributions exhibited significant variations (*p* < 10^−21 for allosteric, *p* = 0.0025 for dual). The predicted distance between residues E762 and G857 was significantly greater when binding to allosteric inhibitors compared to orthosteric inhibitors. This distance can serve as an indicator of the αC-helix-in and αC-helix-out states (Fig. 2d), consistent with previous studies[35,36]. Specifically, the αC-helix-in state represents the active conformation, while allosteric inhibitors tend to bind to αC-helix-out state, which is the inactive conformation. Moreover, the residue-wise distance between the kinase and ligand indicates that the binding sites of allosteric inhibitors are situated in closer proximity to the αC-helix region rather than the ATP-binding sites and the active site (Fig. 2e).

We then used the predicted affinities and distances obtained from Ligand-Transformer to screen a subset of TargetMol containing 9090 compounds in stock ("**Methods**" and Supplementary Data 3). We selected candidates with the criteria that they are predicted to have high binding affinity by all of the 11 models (Table S3). Finally, we obtained 12 candidates (Table S4) with predicted $IC_{50}$ between 1 and 100 nM. These candidates included a compound (brigatinib) previously identified as effective inhibitor targeting $EGFR^{LTC}$ with a reported $IC_{50}$ of 1–38 nM[37]. To our knowledge, among the remaining 11 compounds, C1, C4, C5, C10, and C11 have not been reported to target EGFR, while C7 is known to target EGFR but not to $EGFR^{LTC}$.

We then tested experimentally the inhibitory potency of these 11 candidates, and found six active compounds. Out of these six active compounds, three fell within the predicted IC50 range of 1 to 100 nM, with two of them, C1 and C10, exhibiting high potency, with $IC_{50}$ values of 5.5 and 1.2 nM, respectively (Fig. 2f). C1 is a naphthyridine derivative, which is a non-traditional scaffold for EGFR inhibitors. Based on our predictions, the naphthyridine moiety may be capable of competing with the purine ring of ATP and forming hydrogen bonds with the backbone of M793, thereby stabilizing the ligand within EGFR pocket. C10 and C4 show Tanimoto similarities (based on Morgan fingerprints[38], radius = 2, 2048 bits) of 0.77 and 0.35, respectively, to brigatinib. These compounds exhibit a high degree of pharmacophore overlap, particularly the aniline-pyrimidine scaffold. This structural motif may compete with the purine ring of ATP, occupying the EGFR pocket. Additionally, the pyrimidine moiety can form hydrogen bonds with backbones of M793 and P794, thereby enhancing its binding affinity. C5, C7, and C11 exhibit inhibitory activity against $EGFR^{LTC}$ in the range of 100 nM–10 μM, indicating relatively weak inhibitory potency. C11 is a pyridine compound, which consists of only two aromatic rings. From a structural perspective, it represents a novel EGFR inhibitor. C5 is a pyrimidine-2,4-dione derivative and features a quinoline ring, which is a classic structural motif commonly found in EGFR inhibitors. On the other hand, C7 has a complex structure and a relatively high molecular weight, which does not align with the characteristics of ATP-competitive inhibitors targeting EGFR.

To provide further insights into the binding modes of $EGFR^{LTC}$ inhibitors, we analyzed the predicted distances between residues E762 and G857 for the 11 newly identified candidate compounds (C1 to C11), as shown in Fig. S14a. Most positive inhibitors (C1, C4, C7, C10, and C11) exhibit distances below 5.3 Å, indicating that the αC-helix is in the "in" state. In contrast, C5 displays a slightly longer distance of 5.75 Å, suggesting a possible "out" or intermediate state of the αC-helix. This observation is further illustrated in the distance probability distributions (Fig. S14b), where two distinct peaks, corresponding to the "in" and "out" conformations of the αC-helix, are observed when the αC-helix binds to C5.

In addition to the E762–G857 distances, we analyzed the binding regions of these compounds to assess their interaction with the αC-helix region. As shown in Fig. S14c, all candidates are predicted to bind near the active site, consistent with our objective to identify inhibitors that directly target this region ("**Methods**"). Within this configuration, most positive inhibitors (C1, C4, C7, C10, and C11) are predicted to remain relatively distant from the αC-helix (minimum distance > 6 Å), aligning with the expected orthosteric binding mode. However, C5 is predicted to interact more closely with the αC-helix (minimum distance <5 Å), suggesting again that it may act as a dual inhibitor by interacting with both the active site and the αC-helix region (Fig. S14d). We modeled the complex structures using Protenix (v0.4.4)[39] (Figs. S18–S23). The results confirm our prediction that C1, C4, C7, C10, and C11 predominantly occupy the orthosteric binding pocket, whereas C5 engages both the orthosteric and allosteric pockets. Additionally, when bound to C5, the αC-helix tends to adopt an "out" conformation (Fig. S24).

## Conformational selectivity of ABL kinase inhibitors

Kinases are dynamic, interconverting between different conformational states[40]. Monitoring these transitions and characterizing the conformational states that a kinase populates has proven to be a significant challenge[41]. Kinases typically maintain highly conserved active states to ensure the accurate positioning of essential catalytic regions[35]. In contrast, their inactive states can be distinct among different kinases[42]. Examining how small molecules selectively bind individual kinases could be instrumental in developing inhibitors specifically targeting these unique inactive states[42]. Here, we investigated whether Ligand-Transformer can be used to predict the ligand binding-induced conformational population shift of the ABL kinase, which plays a pivotal role in several signalling pathways, governing crucial cellular processes such as growth, survival, invasion, adhesion, and migration[43,44].

ABL has three major conformational states, one active state (A) and two inactive states ($I_1$ and $I_2$)[41]. To investigate the ability of Ligand-Transformer to capture the change of the conformation ensemble of ABL after binding, we collected 12 inhibitors of this kinase, each with a predominant state of ABL when bound, as determined by nuclear magnetic resonance (NMR) spectroscopy[41]. A total of 60 possible structures of ABL were obtained from the PDB IDs 6XR6, 6XR7, and 6XRG, with 20 structures corresponding to each of the three states A, $I_1$ and $I_2$ (Fig. 3a). We then used the predicted distances between the residues of ABL as constraints to reweight the conformational ensemble consisting of the 60 structures and calculate the population of each conformational state (see "Methods"). Upon binding, the predicted predominant states are in accordance with experimental measurements for 11 of 12 compounds (Fig. 3b). Furthermore, when grouping the inhibitors by their labelled predominant state, we found that each group had a significantly higher population within the corresponding state compared to the other two groups.

We conducted a detailed exploration of the predicted distance distributions that illustrate the distinct conformation ensembles of ABL when bound with various inhibitors. Our analysis (Fig. 4) focuses on the spatial positioning of the αC-helix (residues 299–311), the phosphate-binding loop (P-loop, residues 267–275), and the activation loop (A-loop, residues 400–424) of ABL. These elements are pivotal in differentiating the three states of ABL[41]. We chose four representative residue pairs, selected for their high F-statistics indicating their effectiveness in distinguishing between the three states, which enabled us to investigate the distances between critical protein components.

The ability of Ligand-Transformer to characterise the conformational space of the protein target and its ligand in their complex state can be appreciated by analyzing the distance between V308 and F401, which represents the distance between the αC-helix and the DFG motif, comprising residues D400, F401, and G402, at the start of the activation loop (Fig. 5). The increased distance in states A, $I_1$, and $I_2$ is notable, reflecting the DFG-out conformation in states $I_1$ and $I_2$, and the DFG-in state in state A[41]. The distance between residues V275 and F401 represents the gap between the DFG motif and the P-loop (Fig. S15). This distance is significantly reduced in state $I_2$, which adopts a P-loop stretched conformation, as opposed to the P-loop kinked conformation in states A and $I_1$[41]. Predictions from Ligand-Transformer corroborate this trend. The distance between residues H380 and G402 represents the span from the active site to the activation loop (Fig. S16). In state $I_2$, which adopts an A-loop closed conformation[41], this distance is larger than in the other states, and the measured structures show that the distance in state $I_1$ is slightly shorter than in state A. In state $I_2$, the A-loop-closed conformation results in a shorter distance between L403 and L406, indicating a more compact activation loop (Fig. S17).

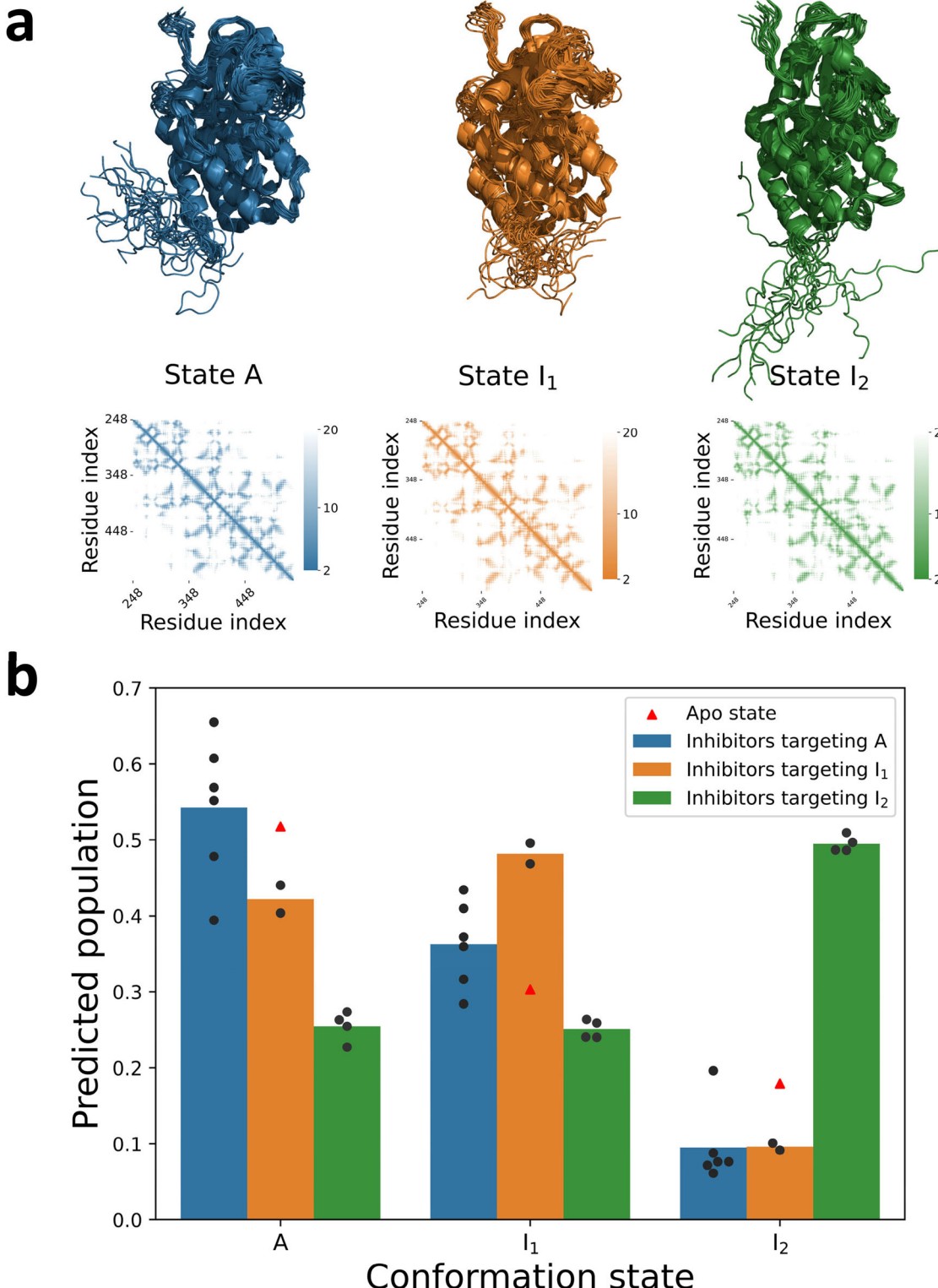

**Fig. 3 | Conformational selectivity of inhibitors targeting ABL kinase. a** Average distance maps and structural ensembles for ABL kinase states A (blue), $I_1$ (orange), and $I_2$ (green), corresponding to PDB IDs 6XR6, 6XR7, and 6XRG, respectively. **b** Conformational selectivity prediction of different inhibitors. Twelve molecules were divided into three groups based on their conformational selectivity as determined in the literature[41]: Group A (blue, $n = 6$), Group $I_1$ (orange, $n = 2$), and Group $I_2$ (green, $n = 4$). The grouped bar graph illustrates the differences in predicted population of binding state conformations of ABL when interacting with inhibitors from different groups. The red triangle represents the population of ABL conformations predicted by AlphaFold2 in the apo state. Population estimations are derived from predicted distance matrices (see "Methods" for details of calculations). The predicted populations of ABL state bound with each individual ligand are listed in Table S5. Source data are provided as a Source Data file.

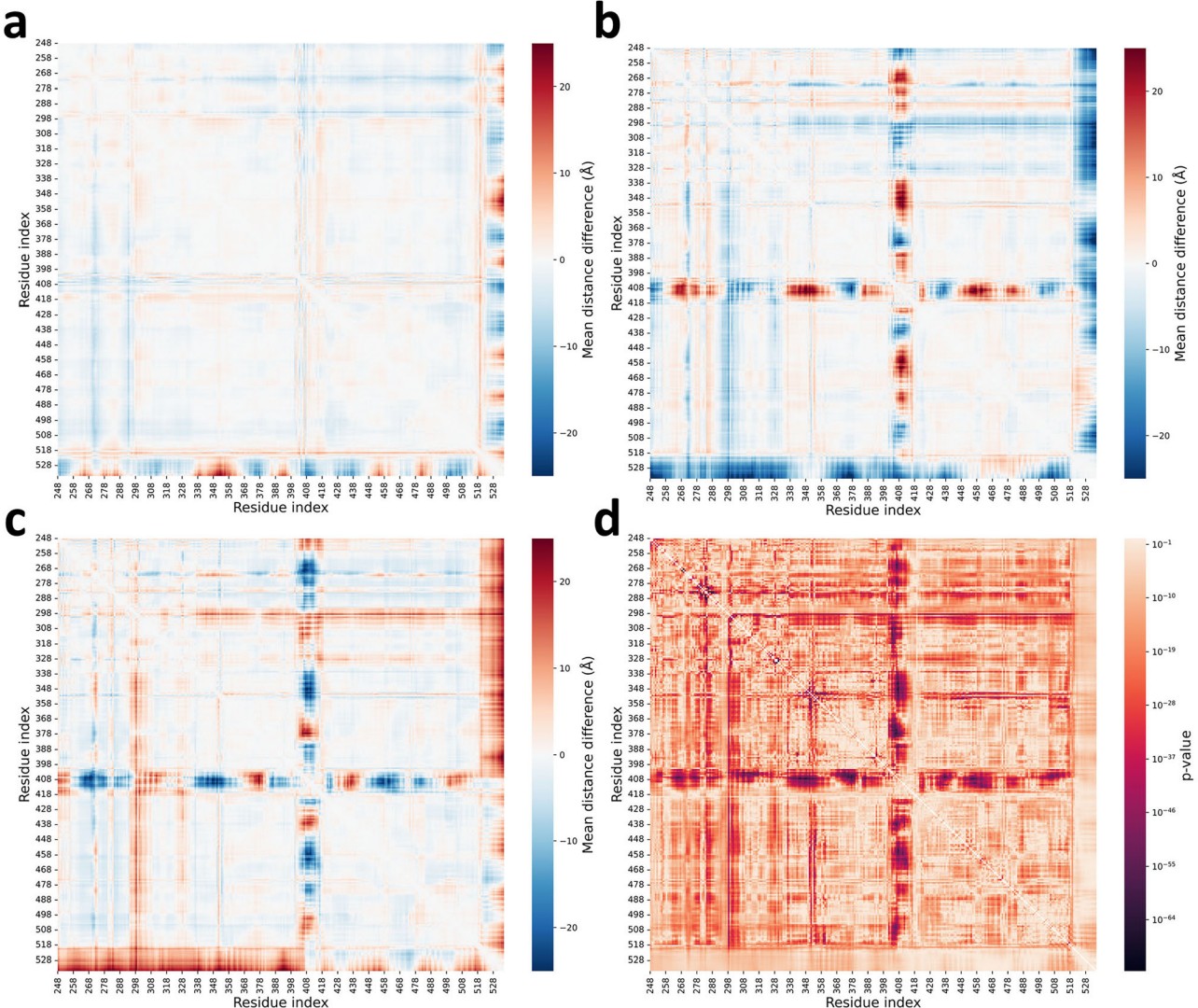

**Fig. 4 | Comparative analysis of distance matrices of the states of ABL. a–c**
Difference distance matrices illustrating the structural variations between the three states of ABL: **a** Difference distance matrix of state $I_1$ minus state A. **b** Difference distance matrix of state $I_2$ minus state A. **c** Difference distance matrix of state $I_1$ minus state $I_2$. **d** F-statistic values derived from the one-way ANOVA to assess variations in distance distributions across each pair of residues among the three states ($n = 20$ per state). This test is two-sided. No adjustments were made for multiple comparisons. The color intensity correlates with the significance of the distance variation, with the accompanying scale indicating $p$-values. Source data are provided as a Source Data file.

These conformational changes are all reflected in the distance predictions of Ligand-Transformer. Specifically, when ABL in the holo state is bound with an inhibitor selective to a specific state, it tends to favor a distance distribution pattern that aligns with the measured distribution of that particular state (Figs. 5 and S15–S17). Moreover, the predicted distance distributions from Ligand-Transformer exhibit a wide range with multiple peaks, differing from the apo state predictions by AlphaFold2. This suggests that Ligand-Transformer can provide information about the structural ensembles of a protein-ligand complex. These conformational changes upon binding are central to the biological regulation of kinase activity. For instance, the active conformation (state A) of ABL is associated with substrate phosphorylation and downstream signaling, while the inactive states ($I_1$ and $I_2$) serve as key regulatory checkpoints. Small molecules that preferentially stabilize specific conformations can either inhibit or activate the kinase, influencing cell fate decisions such as proliferation, apoptosis, and migration. Therefore, the ability to predict conformational population shifts has important implications for understanding drug mechanism of action and for the design of selective inhibitors that avoid off-target effects by exploiting unique features of inactive states. This capability is particularly relevant for kinases like ABL, where mutations can confer drug resistance by altering the accessible conformational landscape.

## Discussion

We described a sequence-based virtual screening method of predicting the conformational space of a target protein and a ligand in their complex state, thus overcoming the limitations of relying on the structures of the binding partners in their free states. In this way, this approach provides the binding affinity and the corresponding binding mode, represented as distance matrices between the target protein and the ligand.

Through comparisons with baseline models[30–32] and ablation experiments, we observed that Ligand-Transformer performs well in affinity predictions (Table S1, Supplementary Discussion). This result can be attributed to the protein and molecular representations provided by pre-trained AlphaFold2[6] and GraphMVP[29], as well as the structural information learned from the distance matrices of protein-ligand complexes during the training.

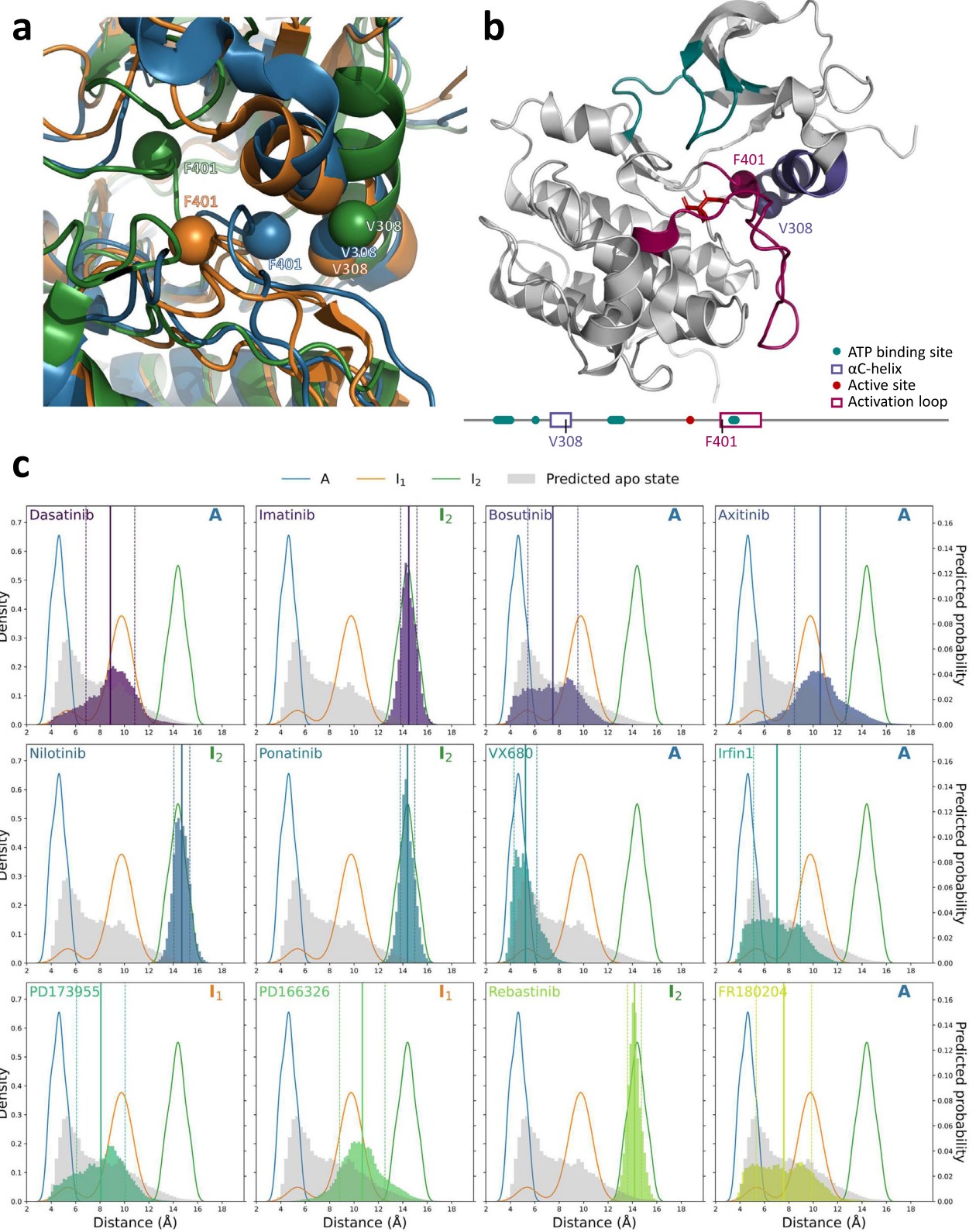

We illustrated the use of Ligand-Transformer by predicting ligands for the mutant EGFR^LTC kinase with a hit rate of over 50% in this particular case (7/12 positive compounds, Table S4), finding novel high-affinity ligands, showing that Ligand-Transformer enables out-of-distribution predictions for specific protein targets. Notably, Ligand-Transformer achieves this accuracy while being over two orders of magnitude faster than state-of-the-art co-folding (Boltz-1)[45] and structure-based docking (Vina-GPU 2.1)[46] methods (Fig. 6).

Next, as the free energy landscape of a protein is crucial in determining its behavior[47–49], we investigated the problem of predicting conformational ensembles, which remains an open problem in the development of next-generation structural prediction algorithms[50].

**Fig. 5 | Analysis of distances between the Cβ atom of residue V308 and the Cβ atom of residue F401 of ABL. a** Structural overlay highlighting the distance between residues V308 and F401 in the conformations of states A (PDB ID 6XR6, blue), $I_1$ (PDB ID 6XR7, orange), and $I_2$ (PDB ID 6XRG, green) of ABL. Residues are represented by spheres at the Cβ atoms, and the proteins are depicted as ribbons. **b** Depiction of the spatial positioning of residues V308 and F401 within the ABL kinase state A (PDB ID 6XR6). The sequence annotation is based on UniProt entry P00519, with the ATP binding site, αC-helix, active site, and activation loop colored in teal, violet, red, and magenta, respectively. The protein backbone is rendered as a cartoon, the active site residues as sticks, and the Cβ atoms of residues V308 and F401 as spheres. **c** Distance distributions between residues V308 and F401 for ABL

in complex with 12 different inhibitors. Kernel density estimations (KDE) plots for the distances from 20 measured structures are shown for state A (blue), state $I_1$ (orange), and state $I_2$ (green). Predicted distance probability distribution of the apo state, derived from AlphaFold2 (AF2), is depicted as grey bars with a bin width of 0.3 Å. The Ligand-Transformer predicted distance probabilities for the 12 inhibitors are displayed as colored bars with a bin width of 0.19 Å. The mean values of predicted distances are plotted as solid lines, with dashed lines representing the standard deviation. The symbol in the upper right corner denotes which conformational state of ABL the inhibitor selectively binds to as determined by NMR analysis[41].

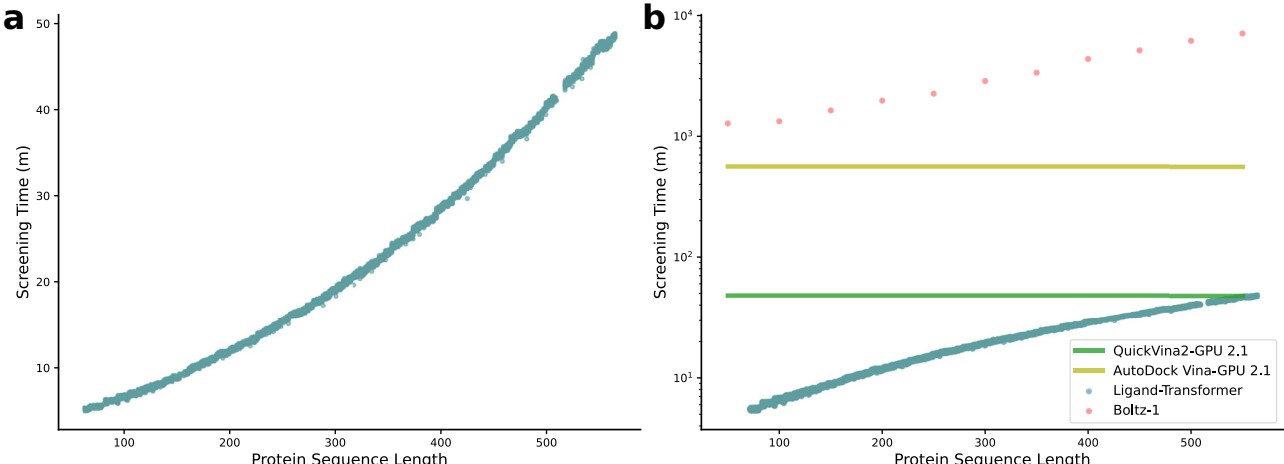

**Fig. 6 | Ligand-Transformer demonstrates greater efficiency than other state-of-the-art docking methods.** Estimated time to screen 10,000 molecules using Ligand-Transformer (**a**) and other popular docking methods (**b**). Observe that Ligand-Transformer is two orders of magnitude faster than Boltz-1.

In this context, Ligand-Transformer offers an approach towards understanding the impact of ligand binding on conformational ensembles using predicted distance matrices. Specifically, the rational design of allosteric modulators of kinases, particularly type-4 inhibitors[42] and activators[51], often encounters challenges with the traditional lock-and-key paradigm. Our methodology proposes a potential solution, enabling the design of ligands that modulate kinase activity by influencing the population distribution across various conformational states.

The results that we presented indicate that Ligand-Transformer can accurately predict the interactions of small molecules with proteins, including binding affinities and population shifts, thus helping understand molecular mechanisms, and offering a tool to replace high-throughput experimental assays in the initial steps in drug design pipelines.

## Methods

### Protein representations generated by AlphaFold2

We utilized AlphaFold2, a pre-trained model to extract protein representations[6]. The protein representation consists of three key components. Firstly, we generated the single sequence representation $\{\mathbf{f}_i^{\text{msa}}\}$, where $\mathbf{f}_i^{\text{msa}} \in \mathbb{R}^{c_{\text{msa}}}$, $c_{\text{msa}} = 384$, and $i \in \{1...N_{\text{res}}\}$. This representation is derived by linearly projecting the first row of the multiple sequence alignment (MSA) representation. The MSA representation is the output of the final layer of the Evoformer and serves as the input to the structure module in AlphaFold2[6]. Secondly, we established pair representations $\{\mathbf{f}_{ij}^{\text{pair}}\}$ for each pair of residues $i$ and $j$, where $\mathbf{f}_{ij}^{\text{pair}} \in \mathbb{R}^{c_{\text{pair}}}$, $c_{\text{pair}} = 128$. The pair representations $\{\mathbf{f}_{ij}^{\text{pair}}\}$ are also outputs of the Evoformer and the input to the structure module in AlphaFold2 and are used in predicting inter-residue distances in AlphaFold2. Lastly, the structure representation $\{\mathbf{f}_i^{\text{struc}}\}$ was obtained from the final layer of the structure module in AlphaFold2, where $\mathbf{f}_i^{\text{struc}} \in \mathbb{R}^{c_{\text{struc}}}$,

$c_{\text{struc}} = 384$, and $i \in \{1...N_{\text{res}}\}$. This structure representation is employed in AlphaFold2 for predicting side-chain dihedral angles and model confidence prediction. The MSA searching was conducted by MMseqs2 (default setting) on BFD/MGnify and Uniclust30 (2021_03). No structural template was fed during the prediction. Model 1.1.1 of AlphaFold2 was used for the inference, and no structural template was used.

### Ligand representations generated by GraphMVP

For obtaining a stereospecific molecular geometry representation, we used GraphMVP, a pre-trained molecule encoder that transforms 2D ligands into graph representations[29]. GraphMVP is trained via self-supervised learning (SSL) using auxiliary tasks. Here, both the input atoms and chemical bonds underwent a one-hot encoding process before being input into GraphMVP. The outputs generated include the atom representation, denoted as $\{\mathbf{f}_k^{\text{atom}}\}$, where $\mathbf{f}_k^{\text{atom}} \in \mathbb{R}^{c_{\text{atom}}}$, $c_{\text{atom}} = 100$, and $k \in \{1...N_{\text{atom}}\}$, and the bond representation, represented as $\{\mathbf{f}_{kl}^{\text{bond}}\}$, where $\mathbf{f}_{kl}^{\text{bond}} \in \mathbb{R}^{c_{\text{bond}}}$, $c_{\text{bond}} = 100$, for each chemical bond formed between atom $k$ and atom $l$.

### Datasets

**PDBbind2020-subset.** We utilized the publicly accessible PDBbind v2020 dataset[37], which provides the structures of 19,443 protein-ligand complexes along with their experimentally measured binding parameters. We eliminated structures with more than one ligand and those whose ligand structures could not be read by RDKit (http://www.rdkit.org/). Additionally, to enhance the training efficiency, we initially limited the dataset to complexes where the ligand atom count was 128 or fewer, and the protein sequence length was restricted to 384 amino acids or fewer. For sequences exceeding 384 amino acids in length, we applied a truncation strategy (see Supplementary Methods and Discussion) by eliminating domains that

were distant from the binding site. This resulted in a subset dataset (PDBbind2020-subset) consisting of 13,420 data points. We subsequently randomly divided the dataset into training, validation, and testing sets, containing 10,375, 640, and 936 data points, respectively (see Supplementary Data 1). During model training, we initially conducted warm-up training with 4480 complexes (PDBbind_4k) that only included single chains with lengths less than 384. Following this, we proceeded to train with the complete training set (PDBbind_10k) of 10,375 data points (see Supplementary Methods and Discussion for training details).

For activity data, the PDBbind dataset includes dissociation constant ($K_d$), half maximal inhibitory concentration (IC$_{50}$), and inhibition constant ($K_i$) measurements. As prediction labels, we took the negative logarithm of these values (in molar units), resulting in the p$K_d$, pIC$_{50}$, and p$K_i$ values, respectively. We considered $K_i$ and $K_d$ measurements without distinction to train Ligand-Transformer to predict $K_d$ values. The IC$_{50}$ data are assay-specific and only comparable under certain conditions. Based on literature research[52], augmenting mixed public IC$_{50}$ data with public $K_i$ data does not deteriorate the quality of the mixed IC$_{50}$ data if the $K_i$ is corrected by an offset. Therefore, we did not explicitly distinguish among the three types of activity data. Instead, we corrected all IC$_{50}$ values by dividing them by a factor 2.3. That is, we added 0.35 log units to all pIC$_{50}$ values to obtain their p$K_d$ equivalents for use as prediction labels. Ligand-Transformer can thus predict $K_d$, IC$_{50}$, and $K_i$ values.

For the distance data, the ground truth distances are computed from the PDB files using Biopython. As the distance prediction has a lower limit of 1 Å and an upper limit of 20 Å, all labels are truncated within this range when calculating the absolute error between the predicted distances and true distances.

**EGFR$^{LTC}$-290.** We collected 400 inhibitors targeting the L858R/T790M/C797S triple-mutant EGFR from ChEMBL[53] and SciFinder (as of November 2022). These inhibitors were obtained from about 90 literature sources, and all data were double-checked with the original publications to ensure some important information, e.g., tested with human protein, and L858R/T790M/C797S triple-mutant EGFR form. For duplicates, we retained the one with higher bioactivity, resulting in a final set of 290 inhibitors with IC$_{50}$ values ranging from 0.08 nM to 150 μM. We classified the inhibitors into orthosteric inhibitors, allosteric inhibitors, or those that bind to both sites, based on information from their originally published literature.

**TargetMol library for screening EGFR$^{LTC}$ inhibitors.** The library used for virtual screening for EGFR$^{LTC}$ inhibitors in this study contained a total of 9090 compounds, all sourced from TargetMol. This collection encompassed 2040 approved drugs, 5370 bioactive compounds, and 1680 natural compounds. Following the deduplication process, the final count of unique compounds in the library was reduced to 5600, as detailed in Supplementary Data 3.

## Screening strategy of EGFR$^{LTC}$ inhibitors
When selecting candidate ligands from the TargetMol library for binding to EGFR$^{LTC}$, we first considered the predicted binding activity. We normalized the predicted affinities from 11 models, including the base model (training with hyperparameters described in Table S3) and fine-tuned models (Model FT1 to FT10), and defined an overall affinity score as the minimal predicted affinity (normalized) among the 11 models (see Supplementary Methods and Discussion). In screening the candidate ligands, we also took into account the binding location of the ligands. We calculated the predicted distances of the ligand to residues K745, E762, D855, the three functional residues of EGFR, and normalized these distances to obtain a distance score. We screened molecules based on the following criteria: overall affinity score > 0.3 (i.e., top 50 results) or affinity score of any model rank <10; and all distance scores <0.5, to ensure that the

binding location of the small molecule is not too far from the target region. Following these criteria, we obtained 12 candidates (Table S4).

## Reweighting state populations of the ABL kinase
ABL exists in three different states, with state A corresponding to PDB ID 6XR6, state I$_1$ corresponding to PDB ID 6XR7, and state I$_2$ corresponding to PDB ID 6XRG. Each PDB file contains a set of $N_{conf} = 20$ conformations. We use the average residue distance among the 20 conformations in each state as the distance matrix $\{d_{ij}^s\}$. Specifically, $d_{ij}^s = \frac{1}{N_{conf}} \sum_{n=1}^{N_{conf}} d_{ij}^{s(n)}$, where state $s \in \{A, I_1, I_2\}$, and $i,j \in \{1, \ldots, N_{res} = 287\}$. The conformational ensemble of the ABL kinase can be simplified and represented by the weights $\mathbf{w} = [w^A, w^{I_1}, w^{I_2}]$ of the three states. Consequently, the distance matrix $\{\bar{d}_{ij}(\mathbf{w})\}$ of the conformational ensemble can be expressed as a weighted average: $\bar{d}_{ij}(\mathbf{w}) = \sum_s d_{ij}^s w^s$. During the reweighting process, we optimize the weights $\mathbf{w}$ to minimize the mean squared error (MSE) between the protein distance matrix $\{\hat{d}_{ij}\}$ predicted by Ligand-Transformer and the distance matrix $\{\bar{d}_{ij}(\mathbf{w})\}$ of the conformational ensemble. The optimized weights $\mathbf{w}^*$ are considered to represent the conformational population corresponding to the Ligand-Transformer predicted distance matrix. Specifically, we use Sequential Least Squares Programming (SLSQP) from SciPy to solve the following optimization problem with constraints and bounds:

$$
\begin{aligned}
&\min_{\mathbf{w}} MSE(\mathbf{w}) = \frac{1}{N_{res}^2} \sum_{(i,j)} \left( \bar{d}_{ij}(\mathbf{w}) - \hat{d}_{ij} \right)^2 \\
&\text{s.t.} \, 0 < w^s < 1 \, \text{for} \, s \in \{A, I_1, I_2\} \\
&\sum_s w^s = 1
\end{aligned}
\tag{1}
$$

## Baselines
**HAC-Net.** We obtained the code for HAC-Net from the official repository at https://github.com/gregory-kyro/HAC-Net, and used their pre-processed input HDF files for the PDBbind2020 dataset. We adjusted the dataset split to align with the division we utilized for training and testing. The training process and hyperparameters adhered to their default settings.

**TankBind.** We sourced the code for TankBind from https://github.com/luwei0917/TankBind. Using their dataset construction script, we constructed a dataset split for PDBbind2020 consistent with what we used in our study. We adhered to their default training settings to retrain their model on our version of the dataset.

**MONN.** We obtained the code from https://github.com/lishuya17/MONN. Their dataset pre-processing discards complexes with molecules like nucleic acids and polypeptides. Therefore, we made certain modifications to their data handling code to accommodate all the data we required. We rebuilt the dataset to align with the split we used in our paper and retrained their model using their default parameters.

## In vitro EGFR$^{LTC}$ inhibition assays
The HTRF KINASE-TK assay kit (Cat#62TK0PEJ) was purchased from PerkinElmer for evaluating compound inhibition against EGFR L858R/T790M/C797S kinase. Compounds were serially diluted in DMSO to achieve a final assay concentration 200-fold lower than the detection concentration, maintaining a consistent final DMSO content of 0.5% in the assay system. A total of 25 nL of each compound was transferred to a 384-well reaction plate (Greiner, Cat#784075) using an Echo655 acoustic dispenser. The EGFR L858R/T790M/C797S kinase was used at a working concentration of 0.7 nM, chosen within the linear range where the reaction rate remains constant to ensure accurate activity

measurements. The kinase solution was prepared in 1× kinase reaction buffer (comprising 5 mM $MgCl_2$, 1 mM DTT, and 1 mM $MnCl_2$), and 2.5 μL of this solution was added to each well. The plate was centrifuged at 140 g for 1 min and incubated at 25 °C for 10 min. To initiate the reaction, a substrate-ATP mixture was prepared in the same kinase reaction buffer, containing TK substrate at a saturating working concentration of 1 μM and ATP at 0.2 μM, a concentration approximately equal to the $K_m$ value. This ATP level was selected in accordance with the Cheng-Prusoff equation (where ATP = $K_m$, $IC_{50} = 2K_i$) to accurately reflect the inhibitory potency of test compounds. Then, 2.5 μL of the substrate-ATP mixture was added to each well, followed by centrifugation at 140 g for 1 min. The plate was sealed and incubated at 25 °C for 50 min. Subsequently, a 2× detection reagent mixture containing XL665 and anti-phospho-tyrosine antibody in detection buffer was prepared, and 5 μL of this solution was added to each well. The plate was centrifuged again (140 g, 1 min) and incubated at 25 °C for 60 min.

Fluorescence emissions at 620 nm (Cryptate) and 665 nm (XL665) were measured using a microplate reader. Each assay condition was tested in duplicate. The percentage of kinase inhibition induced by the compounds was quantified using the equation:

$$\%\text{Inhibition} = 100\% - \left[ \frac{(\text{compound response} - \text{positive control})}{(\text{negative control} - \text{positive control})} \right] \times 100 \quad (2)$$

To evaluate the potency of the inhibitors, $IC_{50}$ values and dose-response curves were generated using GraphPad Prism 7.0 software. This was achieved by fitting the calculated percentage inhibition and the logarithm of the compound concentrations to a variable slope (four-parameter) nonlinear regression model. The model is expressed by the equation:

$$Y = \text{Bottom} + \frac{(\text{Top} - \text{Bottom})}{1 + 10^{(\text{LogIC}_{50} - X) \times \text{Hill slope}}} \quad (3)$$

where X denotes the log of the inhibitor concentration and Y represents the percentage inhibition. Furthermore, the percentage of enzymatic activity in the presence of inhibitors, expressed as % Control, is derived by subtracting the % Inhibition from 100%.

### Screening efficiency
To evaluate the computational efficiency of Ligand-Transformer compared to other state-of-the-art docking methods, we estimated the runtime required for a typical virtual screening task of 10,000 molecules against protein targets of varying lengths. For Ligand-Transformer, we screened a subset of FDA-approved drugs ($n = 2500$) against all proteins in the human proteome with fewer than 565 residues, while for Boltz-1 we evaluated a reduced set of 11 proteins ranging from 50 to 550 amino acids in length (sampled every 50 amino acids) with 100 molecules due to its substantial computational cost; in both cases, we extrapolated these results to estimate the time required for screening 10,000 molecules. For traditional docking methods such as Vina, runtime depends primarily on docking box size rather than protein length. Therefore, we used the average runtime across proteins in the AutoDock-GPU 140 benchmark[46] set as an estimate of typical performance, treating this as a constant independent of protein length.

### Reporting summary
Further information on research design is available in the Nature Portfolio Reporting Summary linked to this article.

## Data availability
All structural models predicted by Protenix (v0.4.4, https://protenix-server.com) are available at the ModelArchive Database (https://modelarchive.org) with the identifiers ma-2zc19, ma-e6x11, ma-sjtz3, ma-7o3rc, ma-jzm4l, and ma-3jjn5. Additional data generated in this study are provided in the Supplementary Information. Source data are provided with this paper.

## Code availability
The code used to develop the model, perform the analyses, and generate the results in this study is Ligand-Transformer (v0.1.0, https://github.com/zshengyu14/LigandTransformer), publicly available and free for academic use under the MIT License (snapshot archived at Zenodo, https://doi.org/10.5281/zenodo.15467622). Third-party software used includes ColabFold (v1.3.0, https://github.com/sokrypton/ColabFold)—MIT License; GraphMVP (https://github.com/chao1224/GraphMVP)—MIT License; HAC-Net (v1.4.2, https://github.com/gregory-kyro/HAC-Net)—MIT License; TankBind (v0.5.0, https://github.com/luwei0917/TankBind)—MIT License; and MONN—available for non-commercial research use only. Original license headers and copyright statements have been retained in all source files.

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

## Acknowledgements

We would like to acknowledge funding from UKRI (10059436 and 10061100, M.V.).

## Author contributions

S.Z., S.P.O., and M.V. conceived the project. S.Z., D.H., R.I.H., and S.P.O., performed the computational work. D.H., R.I.H., and Y.Q. performed the experimental work. M.V. and A.Y. funded the project. All authors analyzed data and wrote the article.

## Competing interests

The authors declare no competing interests.
