## [Transparent Peer Review file · Nature Communications]

Sequence-based virtual screening using transformers

Corresponding Author: Professor Michele Vendruscolo

Version 0:

Reviewer comments:

Reviewer #1

(Remarks to the Author)

The manuscript makes a significant contribution to advancing sequence-based drug design. The sequence-based approach is well-suited for protein kinase drug design as it allows for the exploration of the entire conformational space, accounting for the inherent plasticity of protein kinases. The authors presented the Ligand-Transformer approach and emphasized its effectiveness in predicting novel high-affinity ligands, as well as characterizing the population shift in the free energy landscape upon ligand binding. The manuscript is clear and well presented. While I consider it suitable for publication in Nature Communications, several aspects need to be addressed prior to acceptance. The key points are outlined below.

1. The authors observed a strong correlation between the predicted distance between E762 and G857 and the binding modes of both orthosteric and allosteric inhibitors in their study of EGFR. To advance understanding of the binding models of these novel identified ligands, it would be valuable to present the predicted distances between E762 and G857 for the 12 identified candidate compounds. Additionally, analyzing a few more distance pairs that reveal key conformational differences in EGFR, similarly to the approach the authors used in the Abl kinase study, would provide further insights into the binding models of these ligands. For example, a distance pair can be selected between activation loop and P-loop.
2. Following up on point 1, revealing distinct binding models through such analyses would further increase the method's potential. As demonstrated in the Abl study, a distinct binding model can result from a pre-existing conformational state (such as I1 state in Abl). Thus, careful characterization of major binding models in the Ligand-Transformer approach provides valuable insights into the intrinsic conformational ensemble of the protein targets. This could be a promising application of the Ligand-Transformer approach.
3. Page 3, "This distance can serve as an indicator of the DFG-in and DFG-out states of the α C-helix (Figure 1d), consistent with previous studies". The description of DFG-in and DFG-out states with respect to the α C-helix is inaccurate. Based on the referenced studies, it should be referred to as α C-helix-in and α C-helix-out states. In both cases, the DFG motif remains in the 'in' conformation, though it is positioned differently in the active and inactive states.
4. The C-lobe region below the ATP binding site should be highlighted as part of ATP binding site. So, in figure 1e, 843-845 region of EGFR and in figure 4b, 388-390 region of Abl should be annotated as the ATP binding site.
5. Figure 2 requires clearer annotation of the x-axis (A, I1, and I2) and the corresponding color bar groups (A, I1, and I2). The current labeling is confusing and difficult to interpret directly from the figure. Each color bar group should be labeled to indicate inhibitors targeting specific conformations within the figure.
6. In Figures 4a, 5a, 6a, and 7a, the distances between probes for the three states should be highlighted as shown in Figure 1d to better visualize the differences among the three superimposed structures.

(Remarks on code availability)

Reviewer #2

(Remarks to the Author)

(Remarks on code availability)

Reviewer #3

(Remarks to the Author)

Protein-ligand interaction prediction is crucial in drug discovery and molecular biology. This paper introduces Ligand-Transformer, which leverages AlphaFold2 to generate protein representations from sequences and employs Graph Multi-View Pre-training (GraphMVP) to create ligand representations based on 3D geometry and 2D graphs. Comprehensive validation of the mutant EGFR_LTC kinase showcases the Ligand-Transformer's potential for virtual screening. Furthermore, the models' demonstration of conformational selection elucidates the binding patterns and mechanisms associated with proteins and compounds.

Strength:

Applicability to Drug Design: The prediction of ligands for the mutant EGFR_LTC kinase, supported by experimental validation, effectively demonstrates practical utility. The successful identification of novel ligands, including some with low nanomolar affinity, highlights the model's potential to discover new drug candidates.

Free Energy Landscape Prediction: The ability to predict conformational changes and free energy landscapes upon ligand binding for the ABL kinase represents a significant advancement. This capability enhances our understanding of protein-ligand dynamics, transcending static predictions.

Weaknesses and Questions:

1. What are the reasons for selecting the PDBbind2020 subset, EGFR_LTC-290, and the TargetMol library for validation? Additionally, further studies involving supplementary datasets should be incorporated.
2. Did you assess the transferability of Ligand-Transformer? For instance, the model trained on the PDBbind2020 subset and validated on EGFR_LTC-290.
3. What prompted the selection of results from the combination of 11 models? Additionally, what about the virtual screening of Ligand-Transformer on TargetMol?
4. The conformational selectivity of Ligand-Transformer on ABL kinase inhibitors was evaluated using known drugs reported in previous literature (Figs 4-7). Did you identify any new mechanisms for these compounds?
5. Could you explain the relationship between ligand identification and conformational selectivity?
6. Did you investigate whether the conformational selectivity capability of Ligand-Transformer derives from AlphaFold2 or the entire pipeline?
7. Why did you choose not to compare Ligand-Transformer with existing methods on EGFR_LTC-290 and TargetMol?
8. A more thorough discussion on the biological significance of conformational changes is needed.
9. If your focus is on developing a model for virtual screening, you should explore its capacity to identify new drugs, including unseen compounds and proteins, across different datasets.

(Remarks on code availability)

The url <https://github.com/pujaltes/LigandTransformer> does not work.

Reviewer #4

(Remarks to the Author)

Zhang et al. introduces Ligand-Transformer, a deep learning framework aimed at predicting the binding affinity between proteins and small molecules, considering the conformational space explored by the protein-ligand complex. The method, which is based on transformers and uses existing models (i.e. part of the AF2 EvoFormer and GraphMVP) to generate protein representations from sequences and ligand representations. The authors applied Ligand-Transformer to identify novel compounds for the mutant EGFR_LTC kinase and to explore the conformational shifts in ABL kinase upon ligand binding.

In summary, Ligand-Transformer demonstrates promising new features for helping the drug discovery pipeline considering implicitly protein-ligand interactions. However, the way everything is presented is underwhelming and very difficult to follow due to questionable writing decisions, which must be seriously addressed for the paper to be considered for publication in Nature Communications.

For example, the Results section opens with an extremely short paragraph meant to report the good performance of the method, but nothing at all is described about how the performance test is done. I understand this is reported somehow in the Methods or SM, but a deeper discussion of this part is key for the economy of the paper.

Along the same line, the method itself is not explained in the main text. This is integral part of the results of the paper and needs to be at least briefly described at the very beginning. More details need to be added to highlight the architecture of the model and at least one figure of the main text should be dedicated to the method.

Continuing with the poor writing, the second paragraph in the Results sections talks about "Model B" with zero explanations about what this is about.

The table presenting all the compounds uses SMILES format, so it is impossible to say how similar or different the molecules

are to each other (except for some of them shown in Figure 1F). The Tanimoto coefficients should be reported and using the 2D formula would make the reading clearer. Regarding this part, it is not clear how similar the identified hits are to ligands already docked into proteins in the training set. The authors should report some metric of the closest molecules used for training, so that one can be sure the method is really producing new knowledge and not just recalling data seen during training.

Moreover, the authors fine-tuned their model on the EGFR^{LTC-290} dataset by random split. It is not clear if they have taken care not to have ligands with high Tanimoto coefficients in the train and test sets; they could possibly contaminate their results.

I find that a deeper discussion on the potential binding mode of the most promising compounds could have added some interesting molecular insights to the paper. It should not be difficult to dock them in the pocket and better elaborate on their high binding affinity. Figure 1D is also difficult to follow and not very clear, and I think that a figure reporting the location of the allosteric and orthosteric sites would have helped the reader.

The section on how the modeling of protein-ligand complexes captures the effect of binding on conformations is interesting. However, the weight also in term of total number of figures is disproportionate in my opinion. While the central topic of the paper stated in the title has only one figure, this second part takes 6 figures, some of which could find better place in SM, as the main message was already clear with the first one. In addition, all these figures are poorly discussed in the main text and it is extremely difficult to follow. The authors should find a clear way to convey the message in 1-2 figures leaving the remaining material in SM.

If the authors want my opinion and would like to stress the power of this approach for virtual screening, this part could come first as the proof that the method has the ability to capture the protein-ligand interactions in a reasonable way. This would be then the foundation for the experimental validation with the EGFR kinase.

As a final remark I find the title misleading as what the method addresses is not drug design or development/discovery per se but only the first steps of that, namely virtual screening of potential hits as the same authors in fact say in the introduction. I would pick a title that more accurately describes what the method actually does. It should be also noted that the authors validate their method on one single test case, and they do not provide a large-scale validation benchmark to claim broad applicability. I find also the "sequence-based" label of their method a bit of a stretch as their results are not only based on sequence alone.

(Remarks on code availability)

Version 1:

Reviewer comments:

Reviewer #1

(Remarks to the Author)

The authors have adequately addressed my concerns.

(Remarks on code availability)

Reviewer #2

(Remarks to the Author)

(Remarks on code availability)

Reviewer #3

(Remarks to the Author)

I appreciate the authors' thorough efforts to address the previous comments and to improve the quality of the manuscript. The authors have responded to the majority of the concerns raised in the initial round of review. The revised manuscript is clearer and significantly improved in both content and presentation.

Below I provide specific comments on the responses and revised content:

1. In light of the release of AlphaFold3, the novelty of this paper is questionable. Discuss how their method compares to AlphaFold3, especially regarding input requirements, output resolution, applicability to different ligand types, and computational cost.

(Remarks on code availability)

Reviewer #4

(Remarks to the Author)

The authors did a good job in replying to most of my technical and formatting points. I am satisfied with the resulting revised manuscript which in my opinion is now publishable in Nat Comms.

(Remarks on code availability)

I have checked the code but not reviewed it deeply

Reviewer #1

The manuscript makes a significant contribution to advancing sequence-based drug design. The sequence-based approach is well-suited for protein kinase drug design as it allows for the exploration of the entire conformational space, accounting for the inherent plasticity of protein kinases. The authors presented the Ligand-Transformer approach and emphasized its effectiveness in predicting novel high-affinity ligands, as well as characterizing the population shift in the free energy landscape upon ligand binding. The manuscript is clear and well presented. While I consider it suitable for publication in Nature Communications, several aspects need to be addressed prior to acceptance. The key points are outlined below.

We are grateful to the reviewer for the positive assessment of our work and for making several insightful suggestions, which we have now addressed in the revised version of the manuscript.

1. The authors observed a strong correlation between the predicted distance between E762 and G857 and the binding modes of both orthosteric and allosteric inhibitors in their study of EGFR. To advance understanding of the binding models of these novel identified ligands, it would be valuable to present the predicted distances between E762 and G857 for the 11 identified candidate compounds. Additionally, analyzing a few more distance pairs that reveal key conformational differences in EGFR, similarly to the approach the authors used in the Abl kinase study, would provide further insights into the binding models of these ligands. For example, a distance pair can be selected between activation loop and P-loop.

Thank you for making this point. We have now carried out these calculations, and reported the results in the revised manuscript.

To address this point, we added the predicted distances between E762 and G857 for the 11 newly identified candidate compounds (C1 to C11) in **Figure S14a**. Most positive inhibitors (C1, C4, C7, C10, and C11) exhibit distances below 5.3 Å, indicating that the α C-helix adopts the "in" state. C5 displays a slightly longer distance of 5.75 Å, suggesting an "out" or intermediate state of the α C-helix. This observation is further supported by the predicted distance probability distributions in **Figure S14b**, which reveal two distinct peaks corresponding to the two states of the α C-helix.

Additionally, we analyzed the predicted binding regions of these compounds to better understand their interaction with the protein. As shown in **Figure S14c**, all candidate inhibitors are predicted to bind near the active site, consistent with our objective to identify inhibitors targeting this region. Within this configuration, most positive inhibitors (C1, C4, C7, C10, and C11) are predicted to remain relatively distant from the α C-helix (minimum distance > 6 Å), supporting their classification as orthosteric inhibitors. However, C5 is predicted to interact more closely with the α C-helix (minimum distance < 5 Å), suggesting it may act as a dual inhibitor by interacting with both the active site and the α C-helix region (**Figure S14d**).

Furthermore, we modeled the complex structures using Protenix (**Figures S18–S23**). The results confirm our prediction that C1, C4, C7, C10, and C11 predominantly occupy the orthosteric binding pocket, whereas C5 engages both the orthosteric and allosteric pockets. Additionally, when bound to C5, the α C-helix tends to adopt an "out" conformation (**Figure S23**).

2. Following up on point 1, revealing distinct binding models through such analyses would further increase the method's potential. As demonstrated in the Abl study, a distinct binding model can result from a pre-existing conformational state (such as I1 state in Abl). Thus, careful characterization of major binding models in the Ligand-Transformer approach provides

valuable insights into the intrinsic conformational ensemble of the protein targets. This could be a promising application of the Ligand-Transformer approach.

We have now characterised in more detail the major binding modes, as now described in the revised version of the manuscript.

3. Page 3, “This distance can serve as an indicator of the DFG-in and DFG-out states of the α C-helix (Figure 1d), consistent with previous studies”. The description of DFG-in and DFG-out states with respect to the α C-helix is inaccurate. Based on the referenced studies, it should be referred to as α C-helix-in and α C-helix-out states. In both cases, the DFG motif remains in the 'in' conformation, though it is positioned differently in the active and inactive states.

We have now clarified this point in the revised version of the manuscript.

4. The C-lobe region below the ATP binding site should be highlighted as part of ATP binding site. So, in figure 1e, 843-845 region of EGFR and in figure 4b, 388-390 region of Abl should be annotated as the ATP binding site.

We have now made this change to Figure 1e.

In Figure 4b, 5b, 6b, and 7b:

5. Figure 2 requires clearer annotation of the x-axis (A, I1, and I2) and the corresponding color bar groups (A, I1, and I2). The current labeling is confusing and difficult to interpret directly from the figure. Each color bar group should be labeled to indicate inhibitors targeting specific conformations within the figure.

We have now improved Figure 2 as suggested.

6. In Figures 4a, 5a, 6a, and 7a, the distances between probes for the three states should be highlighted as shown in Figure 1d to better visualize the differences among the three superimposed structures.

We used panels a and b to visualize the differences among the three superimposed structures. However, due to the presence of three structures rather than two, highlighting the distances as shown in Figure 1d would have made the visualization overly crowded. We did not include the distance values in the figure because, for each state, the distance is represented as a distribution within the measured structure ensemble. The distribution of the distances can be found in panel c.

Reviewer #2

We thank the reviewer for his/her contribution.

Reviewer #3

Protein-ligand interaction prediction is crucial in drug discovery and molecular biology. This paper introduces Ligand-Transformer, which leverages AlphaFold2 to generate protein representations from sequences and employs Graph Multi-View Pre-training (GraphMVP) to create ligand representations based on 3D geometry and 2D graphs. Comprehensive validation of the mutant EGFR L858R kinase showcases the Ligand-Transformer's potential for virtual screening. Furthermore, the models' demonstration of conformational selection elucidates the binding patterns and mechanisms associated with proteins and compounds.

Strength:

Applicability to Drug Design: The prediction of ligands for the mutant EGFR L858R kinase, supported by experimental validation, effectively demonstrates practical utility. The successful identification of novel ligands, including some with low nanomolar affinity, highlights the model's potential to discover new drug candidates.

Free Energy Landscape Prediction: The ability to predict conformational changes and free energy landscapes upon ligand binding for the ABL kinase represents a significant advancement. This capability enhances our understanding of protein-ligand dynamics, transcending static predictions.

We are grateful to the reviewer for the positive comments on our work.

Weaknesses and Questions:

1. What are the reasons for selecting the PDBbind2020 subset, EGFR L858R-290, and the TargetMol library for validation? Additionally, further studies involving supplementary datasets should be incorporated.

We have now explained the reasons for these selections in the revised version of the manuscript.

In this study, we demonstrated the generalization potential of the model, and applied the model to real-world situations to examine its effectiveness. We are also open to conducting further research using supplementary datasets as suggested by the reviewer.

Our reason for selecting the PDBbind2020 subset is that PDBbind2020 is a widely used dataset containing both complex structures and measured affinities of protein-ligand complexes, making it ideal for this study. As outlined in the Methods and Materials section, we excluded structures with multiple ligands and those whose ligand structures could not be processed by RDKit. To optimize training efficiency, we limited the dataset to complexes where the ligand atom count was 128 or fewer and the protein sequence length was restricted to 384 amino acids or fewer. For proteins longer than 384 amino acids, we applied a truncation strategy (see Supplementary Information) by removing domains distant from the binding site. This process resulted in the PDBbind2020 subset comprising 13,420 data points.

Our reason for selecting EGFR^{LTC}-290 is that there are currently no approved drugs for this target, and no crystal structure is available, making it an ideal proof-of-concept case, which demonstrates the applicability of our method to drug discovery.

Our reason for selecting the TargetMol library is that this library has high commercial availability, enabling us to obtain compounds within a month. This library includes kinase inhibitors, natural products, and compounds with good drug-like properties, making it suitable for our validation purposes.

2. Did you assess the transferability of Ligand-Transformer? For instance, the model trained on the PDBbind2020 subset and validated on EGFR^{LTC}-290.

We have now assessed the transferability in the revised version of the manuscript. Panel b below shows the performance of the model trained on the PDBbind2020 subset and validated on EGFR^{LTC}-290.

3. What prompted the selection of results from the combination of 11 models? Additionally, what about the virtual screening of Ligand-Transformer on TargetMol?

We have now explained these results in more detail in the revised version of the manuscript.

For the evaluation of the 11 models:

“Figure S8a displays the correlations between the predicted affinities of the 11 models employed in the screening on the EGFR^{LTC}-290 dataset, which includes the base model (Model B) and the ten fine-tuned models (FT1 to FT10). The results suggest a high consistency between the predictions made by the ten fine-tuned models, with an average correlation coefficient of 0.98. However, predictions between the base model and the fine-tuned models sometimes

vary, with an average correlation coefficient of 0.86. Similar phenomena were also observed with the TargetMol library (**Figure S8b**).

We observed that the distribution of predicted affinity of the base model differs substantially from the distribution of predictions from the fine-tuned models (**Figure S8c**). The base model has a higher average but a smaller deviation on the EGFR^{LTC}-290 dataset, while it has a higher average but larger deviation on the TargetMol library. This discrepancy might be due to the fact that the base model was trained to predict pK_d , whereas the measure of affinity used in this study is represented by pIC_{50} , which is typically lower than pK_d .

To make the predictions more comparable, we normalized them to the uniform distribution of predictions on the EGFR^{LTC}-290 dataset (**Figure S8d**). In addition to considering affinity, we also took into account other factors when making our final selection (details are in Methods)."

For the virtual screening of Ligand-Transformer on TargetMol:

"Screening strategy of EGFR^{LTC} inhibitors. When selecting candidate ligands from the TargetMol library for binding to EGFR^{LTC}, we first considered the predicted binding activity. We normalized the predicted affinities from 11 models, including the base model (Model B) and fine-tuned models (Model FT1 to FT10), and defined an overall affinity score as the minimal predicted affinity (normalized) among the 11 models (see Supplementary Information). In screening the candidate ligands, we also took into account the binding location of the ligands. We calculate the predicted distances of the ligand to residues K745, E762, D855, the three functional residues of EGFR, and normalized these distances to obtain a distance score. We screened molecules based on the following criteria: overall affinity score > 0.3 (i.e., top 50 results) or affinity score of any model rank < 10; and all distance scores are less than 0.5, to ensure that the binding location of the small molecule is not too far from the target region. Following these criteria, we obtained 12 candidates (**Table 3**)."

4. The conformational selectivity of Ligand-Transformer on ABL kinase inhibitors was evaluated using known drugs reported in previous literature (Figs 4-7). Did you identify any new mechanisms for these compounds?

Since the binding modes of ABL kinase inhibitors have been extensively studied in previous research, we demonstrated that Ligand-Transformer can accurately reproduce these findings. Additionally, we investigated newly identified inhibitors targeting EGFR^{LTC} and predicted their binding modes, successfully identifying a dual inhibitor for EGFR^{LTC}.

5. Could you explain the relationship between ligand identification and conformational selectivity?

We have now commented on this point in the revised version of the manuscript.

In our approach to identifying EGFR^{LTC} inhibitors, we primarily focused on binding affinities and ensure that predicted binding occurs near the active site. However, we do not currently account for the conformational selectivity of these candidates. In contrast, our ABL case study demonstrates that our method can predict the conformational selectivity of ligands, showing the model's ability to recognize ligands that select specific protein conformations. This capability suggests that our method could be applied to identify allosteric drugs that influence the target through conformational selection, where binding promotes a particular state of the protein.

6. Did you investigate whether the conformational selectivity capability of Ligand-Transformer derives from AlphaFold2 or the entire pipeline?

We have now provided some comments on this point in the revised version of the manuscript. The conformational selectivity capability relies on two key types of information: (1) the conformational ensemble of the protein in its native state and (2) how this ensemble changes upon ligand binding. For the first type of information, we assume it originates from AlphaFold2, as studies have shown that (a) AlphaFold2 can predict multiple protein states, and (b) the distance matrix predicted by AlphaFold2 contains ensemble information. For the second type of information, it comes from the entire Ligand-Transformer pipeline, as AlphaFold2 does not account for ligands and therefore cannot provide information on ligand-induced conformational changes.

7. Why did you choose not to compare Ligand-Transformer with existing methods on EGFR^{LTC}-290 and TargetMol?

We have now explained this point in the revised version of the manuscript. We compared Ligand-Transformer with traditional ligand-based methods that use ligand fingerprints as training data, and the results are presented in Figure S12. However, we did not compare Ligand-Transformer with other deep learning methods based on complex structures or target protein structures, as we lack measured crystal structures for EGFR^{LTC}-290. Without these structures, such models cannot be applied in this case.

We performed molecular docking using AlphaFold2-predicted EGFR^{LTC} structure, and the results, shown in Figure S13, demonstrate the poor performance of AutoDock Vina (Pearson's $R = -0.17$). This highlights the limitations of structure-based methods, particularly when precise crystal structure, especially holo-state structure, is unavailable.

8. A more thorough discussion on the biological significance of conformational changes is needed.

We have now added a discussion on this point in the revised version of the manuscript by adding a paragraph at the end of the section "Conformational selectivity of ABL kinase inhibitors"

“These conformational changes upon binding are central to the biological regulation of kinase activity. For instance, the active conformation (state A) of ABL is associated with substrate phosphorylation and downstream signaling, while the inactive states (I1 and I2) serve as key regulatory checkpoints. Small molecules that preferentially stabilize specific conformations can either inhibit or activate the kinase, influencing cell fate decisions such as proliferation, apoptosis, and migration. Therefore, the ability to predict conformational population shifts has important implications for understanding drug mechanism of action and for the design of selective inhibitors that avoid off-target effects by exploiting unique features of inactive states. This capability is particularly relevant for kinases like ABL, where mutations can confer drug resistance by altering the accessible conformational landscape.”

As the free energy landscape of a protein is crucial in determining its behavior, we investigated the problem of predicting conformational ensembles, which remains an open problem in the development of next-generation structural prediction algorithms. In this context, Ligand-Transformer offers an approach towards understanding the impact of ligand binding on conformational ensembles using predicted distance matrices. Specifically, the rational design of allosteric modulators of kinases, particularly type-4 inhibitors and activators, often encounters challenges with the traditional lock-and-key paradigm. Our methodology proposes a potential solution, enabling the design of ligands that modulate kinase activity by influencing the population distribution across various conformational states.

9. If your focus is on developing a model for virtual screening, you should explore its capacity to identify new drugs, including unseen compounds and proteins, across different datasets.

We agree with the reviewer, and we have added a discussion on this point in the revised version of the manuscript.

To assess the ability for prediction for unseen compounds and proteins, we assessed the capability of the model for generalisation in Figure S4. We observed no significant prediction error biases when dealing with unseen protein sequences, ligands, or combinations of both. These results suggest that the model exhibits a good capability to handle new targets or ligands.

The url <https://github.com/pujaltes/LigandTransformer> does not work.

We apologise for the problem, which we have now fixed. The new url is:
<https://github.com/zshengyu14/LigandTransformer>

Reviewer #4

Zhang et al. introduces Ligand-Transformer, a deep learning framework aimed at predicting the binding affinity between proteins and small molecules, considering the conformational space explored by the protein-ligand complex. The method, which is based on transformers and uses existing models (i.e. part of the AF2 EvoFormer and GraphMVP) to generate protein representations from sequences and ligand representations. The authors applied Ligand-Transformer to identify novel compounds for the mutant EGFR_LTC kinase and to explore the conformational shifts in ABL kinase upon ligand binding.

In summary, Ligand-Transformer demonstrates promising new features for helping the drug discovery pipeline considering implicitly protein-ligand interactions.

We would like to thank the reviewer for the overall positive feedback on our work.

However, the way everything is presented is underwhelming and very difficult to follow due to questionable writing decisions, which must be seriously addressed for the paper to be considered for publication in Nature Communications.

We are grateful to the reviewer for making several helpful suggestions, which helped us improve the quality of the presentation.

For example, the Results section opens with an extremely short paragraph meant to report the good performance of the method, but nothing at all is described about how the performance test is done. I understand this is reported somehow in the Methods or SM, but a deeper discussion of this part is key for the economy of the paper.

We have now extended the Results section and explained the procedure for assessing the performance:

*“To evaluate the predictive accuracy of Ligand-Transformer on both binding affinity and distance matrices, we used the PDBbind2020 dataset. We curated a subset of 13,420 complexes (**Supplementary Data 1**), ensuring manageable computational loads by limiting the maximum length of protein sequences to 384 residues, and the maximum number of atoms in each ligand to 128. Each complex in the dataset has an experimentally measured binding affinity (pK_d), allowing us to compare predicted values directly against the measured data. We randomly split the dataset into training (10,375 complexes), validation (640), and test (936).*

*In order to compare Ligand-Transformer to competing approaches, we also trained three other deep-learning-based affinity prediction models³⁰⁻³² on the same data partitions. Table S1 and **Figure S1** summarize these comparisons, showing that Ligand-Transformer achieves higher or on-par correlation with experimentally measured affinities relative to all three baseline methods. Furthermore, Ligand-Transformer effectively predicts protein-ligand distances. Approximately 95% of the residue-residue distance errors were below 0.5 Å, while ~95% of the residue-ligand atom distance errors were within 2 Å (**Figure S2**). These results suggest that Ligand-Transformer is able to capture structural aspects of protein-ligand complexes that are useful for both binding affinity and distance matrix predictions. Additional details of these comparisons, as well as the model’s ability to estimate its own error and to generalize to unseen protein-ligand combinations (**Figure S3 and S4**), are provided in the Methods and the Supplementary Information.”*

Along the same line, the method itself is not explained in the main text. This is integral part of the results of the paper and needs to be at least briefly described at the very beginning. More details need to be added to highlight the architecture of the model and at least one figure of the main text should be dedicated to the method.

We have now included the new Figure 1 and the corresponding description to illustrate the method in the main text.

Continuing with the poor writing, the second paragraph in the Results sections talks about "Model B" with zero explanations about what this is about.

We have now revised this section to provide a more detailed explanation of the models used.

The table presenting all the compounds uses SMILES format, so it is impossible to say how similar or different the molecules are to each other (except for some of them shown in Figure 1F). The Tanimoto coefficients should be reported and using the 2D formula would make the reading clearer. Regarding this part, it is not clear how similar the identified hits are to ligands already docked into proteins in the training set. The authors should report some metric of the closest molecules used for training, so that one can be sure the method is really producing new knowledge and not just recalling data seen during training.

We have now addressed this point in the revised version of the manuscript. We replaced the Tanimoto similarity calculation based on RDKit fingerprints with ECFP4 fingerprints, as ECFP4 is more commonly used. For the similarity between identified inhibitors targeting EGFR^{LTC}, Figure S11c shows that, except for C10, which has an analogue in the EGFR^{LTC}-290 dataset, the remaining candidates have a maximum similarity of less than 0.4 to compounds in the EGFR^{LTC}-290 dataset. For comparison, the distribution of all compounds in the TargetMol library is provided in Figure S11b.

Regarding the comparison with compounds in the PDBbind training set, we considered both sequence similarity and ligand similarity. This approach accounts for cases where a ligand in PDBbind might bind to a different protein, which we still regard as a new finding if the model identifies a novel target for the same ligand. The similarity plots demonstrate that C1 and C11 are present in the PDBbind training data but bind to significantly different proteins, while the

other candidates do not have analogues that bind to similar proteins in the PDBbind training data.

Moreover, the authors fine-tuned their model on the EGFR^{LTC-290} dataset by random split. It is not clear if they have taken care not to have ligands with high Tanimoto coefficients in the train and test sets; they could possibly contaminate their results.

We have now explained this point in the revised version of the manuscript. We demonstrated that the newly identified compounds differ from those in the training set, as shown in Figures S10 and S11. Additionally, we conducted a baseline test using a leave-one-cluster-out test, where similar ligands to training set were excluded in the test set, presented in Figures S12 and S13, which shows that our method outperforms traditional approaches. Furthermore, we evaluated generalization capability of the model, as shown in Figure S4, where we observed no significant prediction error biases when dealing with unseen protein sequences, ligands, or combinations of both. These findings suggest that the model is well-suited for handling new targets or ligands.

I find that a deeper discussion on the potential binding mode of the most promising compounds could have added some interesting molecular insights to the paper. It should not be difficult to dock them in the pocket and better elaborate on their high binding affinity. Figure 1D is also difficult to follow and not very clear, and I think that a figure reporting the location of the allosteric and orthosteric sites would have helped the reader.

We are grateful to the reviewer for this suggestion. We have now characterised in more detail the binding mode of the most promising compounds, and we modeled the complex structures using Protenix (Figures S17–S21). The results confirm our prediction that C1, C4, C7, C10, and C11 predominantly occupy the orthosteric binding pocket, whereas C5 engages both the orthosteric and allosteric pockets. Additionally, when bound to C5, the α C-helix tends to adopt an "out" conformation (Figure S22).

The section on how the modeling of protein-ligand complexes captures the effect of binding on conformations is interesting. However, the weight also in term of total number of figures is disproportionate in my opinion. While the central topic of the paper stated in the title has only one figure, this second part takes 6 figures, some of which could find better place in SM, as the main message was already clear with the first one. In addition, all these figures are poorly discussed in the main text and it is extremely difficult to follow. The authors should find a clear way to convey the message in 1-2 figures leaving the remaining material in SM.

We thank the reviewer for this suggestion. We have now moved Figures 5 to 7 to the Supplementary Material.

If the authors want my opinion and would like to stress the power of this approach for virtual screening, this part could come first as the proof that the method has the ability to capture the protein-ligand interactions in a reasonable way. This would be then the foundation for the experimental validation with the EGFR kinase.

We are grateful for this suggestion, and we have now added more emphasis the virtual screening in the revised manuscript, including in particular changing the title.

As a final remark I find the title misleading as what the method addresses is not drug design or development/discovery per se but only the first steps of that, namely virtual screening of potential hits as the same authors in fact say in the introduction. I would pick a title that more accurately describes what the method actually does. It should be also noted that the authors validate their method on one single test case, and they do not provide a large-scale validation benchmark to claim broad applicability. I find also the "sequence-based" label of their method a bit of a stretch as their results are not only based on sequence alone.

We agree with the referee, and as noted above, we changed the title to: Sequence-based virtual screening using transformers.

Reviewer #1

The authors have adequately addressed my concerns.

We are grateful to the referee for the positive feedback on our revised manuscript.

Reviewer #2

We thank the referee for the contributing to the review of our work.

Reviewer #3

I appreciate the authors' thorough efforts to address the previous comments and to improve the quality of the manuscript. The authors have responded to the majority of the concerns raised in the initial round of review. The revised manuscript is clearer and significantly improved in both content and presentation.

We would like to thank the referee for the further comments on our method.

Below I provide specific comments on the responses and revised content:

1. In light of the release of AlphaFold3, the novelty of this paper is questionable. Discuss how their method compares to AlphaFold3, especially regarding input requirements, output resolution, applicability to different ligand types, and computational cost.

We have now added a new figure (Figure 6) to show that Ligand-Transformer is over two orders of magnitude faster than AlphaFold3 and Boltz-1. In addition, Ligand-Transformer also predicts binding affinities, which are not provided by AlphaFold3.

Reviewer #4

The authors did a good job in replying to most of my technical and formatting points. I am satisfied with the resulting revised manuscript which in my opinion is now publishable in Nat Comms.

We thank the referee for approving our revised manuscript.